

# DNA from mollusc shell: a valuable and underutilised substrate for genetic analyses

Sara Ferreira[1], Rachael Ashby[2], Gert-Jan Jeunen[1], Kim Rutherford[1], Catherine Collins[1], Erica V. Todd[1] and Neil J. Gemmell[1]

[1] Department of Anatomy, University of Otago, Dunedin, New Zealand
[2] Invermay Agricultural Centre, AgResearch, Dunedin, New Zealand

## ABSTRACT

Mollusc shells are an abundant resource that have been long used to predict the structures of ancient ecological communities, examine evolutionary processes, reconstruct paleoenvironmental conditions, track and predict responses to climatic change, and explore the movement of hominids across the globe. Despite the ubiquity of mollusc shell in many environments, it remains relatively unexplored as a substrate for molecular genetic analysis. Here we undertook a series of experiments using the New Zealand endemic greenshell mussel, *Perna canaliculus,* to explore the utility of fresh, aged, beach-cast and cooked mollusc shell for molecular genetic analyses. We find that reasonable quantities of DNA (0.002–21.48 ng/mg shell) can be derived from aged, beach-cast and cooked mussel shell and that this can routinely provide enough material to undertake PCR analyses of mitochondrial and nuclear gene fragments. Mitochondrial PCR amplification had an average success rate of 96.5% from shell tissue extracted thirteen months after the animal's death. A success rate of 93.75% was obtained for cooked shells. Amplification of nuclear DNA (chitin synthase gene) was less successful (80% success from fresh shells, decreasing to 10% with time, and 75% from cooked shells). Our results demonstrate the promise of mollusc shell as a substrate for genetic analyses targeting both mitochondrial and nuclear genes.

## INTRODUCTION

Molluscs are the most diverse of the marine phyla, with some 85,000 described species (*Chapman, 2009*) that include animals such as clams, slugs and octopuses (*Appeltans et al., 2012*). The characteristic mollusc shell, common to much of the phylum, consists mainly of calcium-carbonate (*Furuhashi et al., 2009*), is highly resilient, and can persist in the environment long after the animal has died (*Vendrasco et al., 2016*). In fossil assemblages, for instance, shell material can be a predictor of ancient community composition (*Kidwell, 2001*), and harbour a rich source of information for exploring mollusc evolution (*Vendrasco et al., 2016*; *Der Sarkissian et al., 2017*; *Coutellec, 2017*) and undertaking paleoenvironmental reconstructions (*Rhoads, 1970*; *Krantz, Williams & Jones, 1987*; *Coutellec, 2017*; *Der Sarkissian et al., 2017*). Likewise, the almost ubiquitous

Corresponding author
Neil J. Gemmell,
neil.gemmell@otago.ac.nz

exploitation of marine mollusc as a coastal food source in many early human communities provides a rich source of information on the lives of those people and the ecology and climate in which they lived (*Balbo et al., 2011*; *Barsh & Murphy, 2008*; *Colonese et al., 2011*).

Whether from natural samples or from samples collected and used by humans, analysis of mollusc shell remains provide important insights into our past. The application of techniques such as radiocarbon dating and the recently developed Amino Acid Racemization (AAR) technique (*Demarchi et al., 2011*) give samples of mollusc shell chronological meaning, while stable isotope analysis enables quantification of nutrient flows and may provide some insights into food web dynamics (*Zanden & Rasmussen, 2001*). Finally, taxonomic analysis of mollusc shell remains provides knowledge of species abundance and community structure, and when coupled with accurate chronology can lead to significant insights into how species, communities and ecosystems respond to a variety of anthropogenic pressures and environmental changes (*Balbo et al., 2011*; *Estevez et al., 2001*).

Species identification based on shell valves often relies on specialised taxonomic knowledge, a scientific skill in continuous decline since the 1950's (*Tautz et al., 2003*; *Hebert & Gregory, 2005*; *Kim & Byrne, 2006*). Unlike DNA from soft tissues, which degrades relatively quickly following the death of the organism, DNA molecules incorporated within mineral matrix such as bone and teeth is largely protected from enzymatic and microbial degradation (*Pääbo et al., 2004*; *Higgins et al., 2015*). The development of techniques that enabled the extraction of trace amounts of DNA from ancient mineralized tissue samples such as bone and teeth (*Kalmár et al., 2000*; *Höss & Pääbo, 1993*; *Rohland & Hofreiter, 2007*) sparked interest in whether other mineralized tissues, such as mollusc shell, might also have utility for DNA analyses (*Doherty, Gosling & Was, 2007*; *Barsh & Murphy, 2008*; *Geist, Wunderlich & Kuehn, 2008*; *Wang et al., 2012*; *Der Sarkissian et al., 2017*).

In bivalves, a group of specialised hematocyte cells called refractive granulocytes has been shown to be involved in the secretion, and active remodelling of calcium carbonate crystals for the formation of the shell (*Mount et al., 2004*; *Li et al., 2016*; *Ivanina et al., 2017*). It is likely that during this process, these cells might be trapped and absorbed in the growing shell, leaving traces of DNA (*Doherty, Gosling & Was, 2007*; *Wang et al., 2012*; *Barsh & Murphy, 2008*; *Der Sarkissian et al., 2017*). These DNA traces provide an opportunity to genetically analyse recently dead individuals, historical specimens, samples from shell middens and even ancient shell samples (*Der Sarkissian et al., 2017*).

Moreover, because living molluscs can quickly repair shell damage (*Fleury et al., 2008*), shell sampling techniques that avoid damaging the underlying mantle epithelia enable non-lethal sampling of valuable animals, such as aquaculture broodstock or those of endangered species (*Wang et al., 2012*).

Despite its broad appeal, the use of mollusc shell as a source of DNA for molecular analysis remains incipient. Here we undertook a series of experiments, using shell remains from the widespread and commercially important bivalve mollusc, the New Zealand greenshell mussel (*Perna canaliculus* (Gmelin 1791)), to develop an effective and reliable method for extracting genomic DNA from mollusc shells. We test the effects of shell age and

treatment (e.g., cooking and weathering) on DNA quality and suitability for downstream analysis by PCR using mitochondrial and nuclear primers.

## MATERIALS & METHODS

For all experiments, live greenshell mussels were acquired from commercial sources in Dunedin, New Zealand. Animals were completely removed from their shells, which were then cleaned with at least three rinses of distilled water before being dried in an incubator at 20 °C overnight. One of the valves was subsequently roughly smashed and a 0.5 g shell sample taken for DNA extraction.

### Ageing shells

For each of 10 individuals, shell samples were taken from the same valve at monthly intervals over 13 months (130 samples in total). DNA extraction was performed immediately after each collection. During the sampling period, shells were individually wrapped in foil and stored at room temperature in a plastic bag to ensure minimal airflow.

### Cooked mussel shells

Sixteen live whole greenshell mussels were divided into four treatment groups, and cooked either by steaming in salted water or cooking over firewood embers, two common ways to cook shellfish, for either five or ten minutes (four mussels per treatment). Shells of cooked mussels were thoroughly cleaned to remove soft tissue, before being rinsed and dried as described above. After drying, large amounts of the periostracum –the outermost layer of the shell - peeled off, particularly around the edges. Shell samples were collected from each animal and immediately processed for DNA extraction.

### Beach-cast shells

Eleven shells (single valves) of *P. canaliculus* were collected from two locations (Dunedin, New Zealand), cleaned and dried as described previously. For each valve there was an attempt to collect samples from (1) the outermost ventral part of the shell; (2) within the area of the pallial sinus, and (3) the umbo of each shell, which would include ligament tissue if present (Fig. 1). Some valves were found broken, and did not allow sampling of all three areas. Given the low DNA yield expected for such shells, we assessed the potential risk of cross-contamination of mussel DNA within our laboratory. An additional sample was collected from 8 of the beach-cast shell valves and used for DNA extraction and amplification in a different research facility (different lab in another building of the University of Otago, Dunedin campus) following the extraction and amplification methods described here, so that the risk of contamination in our own facility could be assessed. Once collected, samples were processed for DNA extraction immediately after collection.

### DNA extraction method

Shell samples were broken into smaller pieces using a mortar and pestle, re-weighed, and placed in a SafeLock two mL tube (Eppendorf) with a stainless steel five mm bead (Qiagen). All samples and two blank tubes (steal bead but no shell) were processed using a TissueLyser II machine (Qiagen) at the maximum frequency setting (30 Hz) for five minutes. DNA
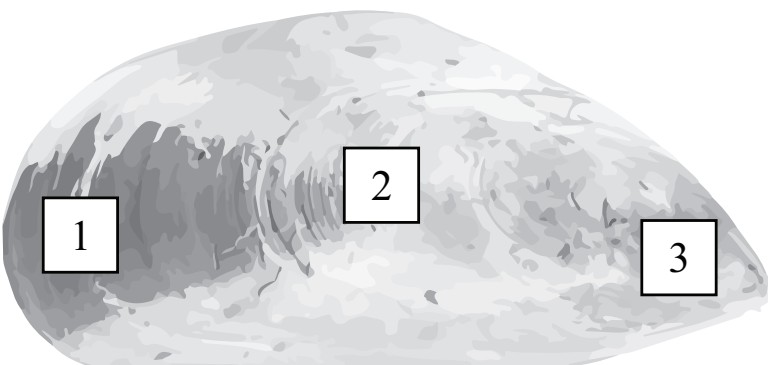

**Figure 1** ***P. canaliculus* shell regions sampled in the beach-cast valves.** (1) ventral region; (2) pallial sinus; (3) umbo. Image credit: Sara Ferreira.

was extracted following a salting-out method (*Gemmell & Akayama, 1996*), with these modifications: (1) lysis incubation was performed overnight; (2) the resulting DNA was resuspended in 100 µL low EDTA TE buffer (10 mM Tris-HCl, 0.1 mM EDTA, pH 8). DNA yields were determined using a Qubit® 2.0 Fluorometer (ThermoFisher Scientific), using the dsDNA HS Assay (ThermoFisher Scientific). DNA (ng)/shell (mg) sample ratios were calculated for each sample. Shell valves from beach-stranded specimens were very brittle, and were processed solely with mortar and pestle. DNA from these samples was eluted in 40 µL low EDTA TE buffer.

## DNA quality assessment by PCR - mtDNA

We designed a set of two mitochondrial DNA (mtDNA) PCR primers to produce amplicons in the range of < 200 to 300 base pairs (bp) (Table 1) using Primer3Plus (*Untergasser et al., 2012*), under the expectation that DNA extracted from bivalve shell material would be degraded and of low yield (*Taberlet, Waits & Luikart, 1999*). Primers were designed using published sequences from the work of *Blair et al. (2006)*, with the exception of one primer which was originally designed by these same authors (*Blair et al., 2006*) (primer COX1 Fwd, see Table 1). Targeted mitochondrial regions were the COX1 gene (primers COX1 Fwd and COX1 Rev, specific to *Perna* spp.) and the NADH4/ ATP8 gene region (primers Pcan Fwd and Pcan Rev, specific to *P. canaliculus* alone) (Table 1). PCR reactions were carried out in a total volume of 30 µL, containing 1x Bioline NH$_4$ Reaction Buffer, 0.1U Bioline BIOTAQ$^{TM}$ DNA Polymerase, 2.75 mM MgCl$_2$ solution, 0.25 µM of each primer, 0.25 µM dNTP's and 1.5 µL DNA. Bovine Serum Albumin (BSA) was also added at a concentration of 10 mM BSA in the PCR using Pcan primers, and 40 mM in the COX1 PCR. Reactions were run in an MJ Research Tetrad PTC-225 Thermal Cycler.

The PCR program for mtDNA primers was 1 min at 96 °C, followed by 35 cycles of 1 min at 96 °C, 30 s at 57 °C and 30 s at 72 °C, with a final elongation step of 5 min at 72 °C. Positive and negative (no template) controls were always included. The positive control was a 1:100 dilution of DNA (1.5 ng/µL) extracted from fresh mantle tissue of one individual sacrificed for the ageing shell experiment, using the DNA extraction method

**Table 1   Primers used to assess DNA quality from shell tissue extracts.**

| Primer | Sequence | Amplicon (bp) | Target Region |
|---|---|---|---|
| **Pcan Fwd** | ACAGACTTCACTTATACACAACAAC | 305 | NADH4 –ATP8 gene region |
| **Pcan Rev** | AGGGCTCTAACTCAATATAACCCC | | |
| **COXI Fwd[a]** | GRGATCCTGTTTTATTTCAGCAYGT | 191 | COX1 gene |
| **COXI Rev** | TGCCCAAACAACACACCCTA | | |
| **Chitin S .Fwd** | CCGTGTCTGTAATGTTGGTCTA | 198 | Chitin Synthase nuclear gene |
| **Chitin S. Rev** | TCTTTGCCATTCGTTCACAC | | |

Notes.

[a]Similar to primer *Pernacox1F* used by *Blair et al. (2006)*.

described by *Gemmell & Akayama (1996)*. PCR product (20 µL) was visualised through a 1.5% agarose gel stained with 1× GelRed (Biotum), together with 5 µL of EasyLadder I DNA ladder (Bioline).

Samples not producing a visible band were re-amplified using template dilutions of 1:5 and 1:10 in water, with the thought that PCR inhibitors compounds might have been co-extracted with biomineralized tissue sourced DNA, and preventing amplification.

Failure to observe a visible band in all PCR attempts led us to designate that particular sample as unsuccessful.

To verify the species specificity of our mitochondrial PCRs, we tested the *Perna canaliculus* DNA positive control template alongside DNA templates from *Perna viridis* (Florida, USA) and *Perna perna* (Haga Haga, South Africa).

## DNA quality assessment by PCR - gDNA

The chitin synthase (CS1) gene previously identified from *Mytilus galloprovincialis* (*Weiss et al., 2006*), was used to test the tractability of obtaining genomic DNA (gDNA) from our shell samples. Using genomic and transcriptomic data already collected by our group for the New Zealand greenshell mussel, a homologue of the chitin synthase gene was identified and its sequence registered on GenBank (*Benson et al., 2012*) with accession reference MT305043. Primers were designed to amplify a 198 bp region within the coding sequence of the *P. canaliculus* chitin synthase gene sequence (Fig. 2). Our analyses suggest this gene is well conserved and apparently single-copy in molluscs. PCRs were performed as described earlier but without BSA, and using an annealing temperature of 53 °C.

To have a sense of the availability of nuclear gDNA throughout the ageing experiment, the chitin synthase PCR was tested with the 0, 1st, 6th and 13th month collections. These primers were also tested in the cooked and beach-cast shell samples. Sample success or failure was explored as described for the mitochondrial amplicons.

## DNA quantification by qPCR

To confirm amplification of mDNA specific to our target species in our extracts, qPCR was also performed in the ageing shell experiment collections at 0, 1, 6 and 13 months, cooked shell experiment and on a subset of the beach-cast shell extracts. qPCR reactions (25 µL) were carried out in triplicates using the SensiMix SYBR Low-Rox kit (Bioline), 1 µl shell

```
MT305043      1  aggtggtggagaatgggaactaccagtatcccttgttcttatttctatcg   50
                 |||||||||.|||||||||..||||||.||||..|.||.||||.|
EF535882.1  3219  aggtggtggtgaatgggaactgccagtatcacttgtattaatatctattg  3268

MT305043     51  gatggtgggagaactacgtgtctggagaatggactgtgtttggaaaaatc  100
                 |.||||||||||||.||.||||||.|||||||.||.||||||||.||.
EF535882.1  3269  gttggtgggagaattatgtgtctggggaatggacagtttttggaaagata  3318

MT305043    101  accataccctttaggcaatggcgttcgattcttcaagatgtacgtgaaac  150
                 ||.||||||.||.|.||||||||||.||.|||.|.||.||.|||||||||
EF535882.1  3319  acgataccattcaagcaatggcgatctattttgcaggacgtacgtgaaac  3368

MT305043    151  gtcttatttgcttattggtccattaaaaaattggattatgcattttgttat  200
                 |||.|||.|.|||.||||||||||||.|||||||||||||||||.||..|.|
EF535882.1  3369  gtcctatgttcttgttggtccattgaaaattggattatgcatatttctct  3418

MT305043    201  caagattgctaactaacaacagcgtttagtactaccagctgctggtgat  250
                 |..||||..||||.|||.||.|.||.||..|.||.|||||||..|.||.|.
EF535882.1  3419  cgcgatttttaaccaacgatagtgtgctggttctaccagcaacaggggaa  3468

MT305043    251  ttcaatgcagcaacaagtgaattttcatcgaaaggagaagaagttggagt  300
                 |||||||||.|.|||||||||.||||||.|||||||||||||||||.|.||
EF535882.1  3469  ttcaatgcaacgacaagtgattttttcgtcgaaggcagaagaagtaggtgt  3518

MT305043    301  ttcgtacagtttgatgtttattcagcttggatgtagcatcatttgcacct  350
                 .||.||.|.||.|||||||||||||||||||.|.|.||.||||||||||.|
EF535882.1  3519  gtcttatagcttaatgttcattcagcttggatctggaataatttgcacat  3568

MT305043    351  atctggccggattagcctgcaaactacacatgcagaaagcagcattcgct  400
                 ||||.||.||..||||.||.|||.||||||||||||||||||||||.||.|.
EF535882.1  3569  atctagctggcctagcttgtaaattacacatgcagaaagcagcctttgca  3618

                                    P.can CS Fwd
MT305043    401  ttacctttaacactagctcctccgtgtctgtaa----tgttg--gtcta  444
                 ||.|||||.|||||.||.|.|.|       ||.|||   .||||  ||||.
EF535882.1  3619  ttgcctttgacacttgcaccac------ctctaaccttagttgtagtctt  3662

MT305043    445  tcttcaatgctcctaccagtttctgccagctcattggcatgttggaggat  494
                 |||.|||||.||.|||||.|||.|||||||.||||||.||||||||||||
EF535882.1  3663  tctgcaatgttcataccaattttgccagcccattggcatattggaggat  3712

MT305043    495  ggttttgtcccgatcttgacttctactctttgttgattcctctgatatgt  544
                 |||||||||.||.||.||.|.||.||.|||...|.||.|||||..||.|||
EF535882.1  3713  ggttttgtccagaactggatttatattccctgctgataccattgatttgt  3762

MT305043    545  gcggtgttgttatggttgtcatacgccatttctgtatctcatatctggtt  594
                 ||.|||.|.|||.|.||||||.||.|.||..|.||.|||||||||||||||
EF535882.1  3763  gcagtgcttttgtggttatcctattcaattacagtatcccatatctggtt  3812

                        P.can CS Rev
MT305043    595  tccacagtgtgaacgaatggcaaagatagaaaa  627
                 .|||||.||||||.||||||||||||.||.|||||
EF535882.1  3813  cccacaatgtgaaagaatggcaaaaattgaaaa  3845
```

**Figure 2** Alignment of the chitin synthase gene in *Perna canaliculus* (Ashby *et al* unpublished) and *Mytilus galloprovincialis* (*Benson et al., 2012*) (GenBank accession number: EF535882.1). Primer sequences highlighted in grey, base differences between *P. canaliculus* and *M. galloprovincialis* in red.

extract sample and Pcan Fwd and Pcan Rev primers (Table 1) (250 nM final concentration). Reactions were run in a QuantStudio 3 Real-Time PCR System (ThermoFisher Scientific), using the following program: initial denaturation with 10 min at 95 °C, followed by 40 cycles of 15 s. at 95 °C, 15 s. at 57 °C and 15 s. at 72 °C, and a melt curve cycle with 15 s. at 95 °C, 1 min at 60 °C and 1 s. at 95 °C. Data collection points were set at every annealing and elongation steps of the cycling program, and at the end of the melt curve cycle. A standard dilution series (undiluted DNA sample and six 1:10 serial dilutions) was made using *P. canaliculus* DNA extracted from fresh mantle tissue of one individual using the DNA extraction method described by (*Gemmell & Akayama, 1996*). DNA amount of

each the standard dilution was quantified with a Qubit® 2.0 Fluorometer (ThermoFisher Scientific), using the dsDNA HS Assay (ThermoFisher Scientific). All six standard dilutions, in addition to undiluted *P. canaliculus* DNA sample and a negative (no template) control, were included in each qPCR run as duplicates. Amplification results were visualised and acquired through the ThermoFisher Connect™ software. A linear regression for the serial dilution series was made for every qPCR run and used correlate amplification results with *P. canaliculus*-specific DNA concentration for every sample tested. Amplification for each sample was determined through its triplicate set results; any triplicate result inconsistent with the other two replicates was discarded. Results with the outcomes "No amplification", or "Undetermined" due to no amplification were quantified as having 0 ng/µl of *P. canaliculus* DNA. Melting curve results for each sample were used to control for the specificity of the amplified products.

## DNA amplicon sequencing and sequence analysis

PCR products of samples showing a visible band of expected size in agarose gels were purified using an AcroPrep™ Advance filter plate (PALL) following the manufacturer's instructions. Purified products were quantified with a NanoDrop 2000 spectrophotometer, and sequenced using an ABI 3730xL DNA Analyser (Applied Biosystems) through the Genetic Analysis Services (GAS) at Otago University. Sequences were checked for taxonomic congruence using BLASTn http://blast.ncbi.nlm.nih.gov (*Benson et al., 2012*) and the results visualised using the BLAST Tree View function. Amplicon sequences were subsequently aligned with the Geneious software against their respective references. Sequences with poor quality (low signal, short length or failed reaction) were re-submitted to GAS, or re-amplified if they were again of poor quality.

Additionally for the COX1 amplicons, all *P. canaliculus* shell sequences were also aligned using MEGA version 7 (*Kumar, Stecher & Tamura, 2016*) together with COX1 sequences from *P. perna* and *P. viridis* we obtained from our COX1 control PCR, and other published COX1 sequences of representative molluscs (AB076920.1 *Tridacna crocea*; AY260822.1 *Antalis sp.*; GU802411.1 *Tonicella marmorea*; HM431980.1 *Octopus rubescens*; HM862494.1 *Limacina helicina*; HM884239.1 *Mercenaria mercenaria*; HM884246.1 *Modiolus modiolus*; JF862383.1 *Margarites costalis*; JF912374.1 *Mytillus galloprovincialis*; KF643468.1 *Onchidoris muricata*; KF644043.1 *Mytilus trossulus*; KF644180.1 *Littorina littorea*; KF644349.1 *Crepidula williamsi*; KM198004.1 *Mytilus coruscus*). A phylogenetic tree was constructed using Maximum Likelihood approaches with MEGA version 7 (*Kumar, Stecher & Tamura, 2016*) using KX121879.1 *Capitella capitata* (annelid) as outgroup (Fig. 3).

## RESULTS

### DNA extraction and quantification

Qubit measured DNA yields ranged from 0.01 to 88.4 ng DNA/µl (0.002 to 21.48ng DNA/mg shell) (Fig. 4), with an overall average of 13.3 ng/µl. In the shell ageing experiment, DNA yields were highly variable in the first collection at 0 months (Fig. 4), and included the sample with the highest yield (88.4ng/µl or 21.48 ng/ mg shell). Cooked shells included the

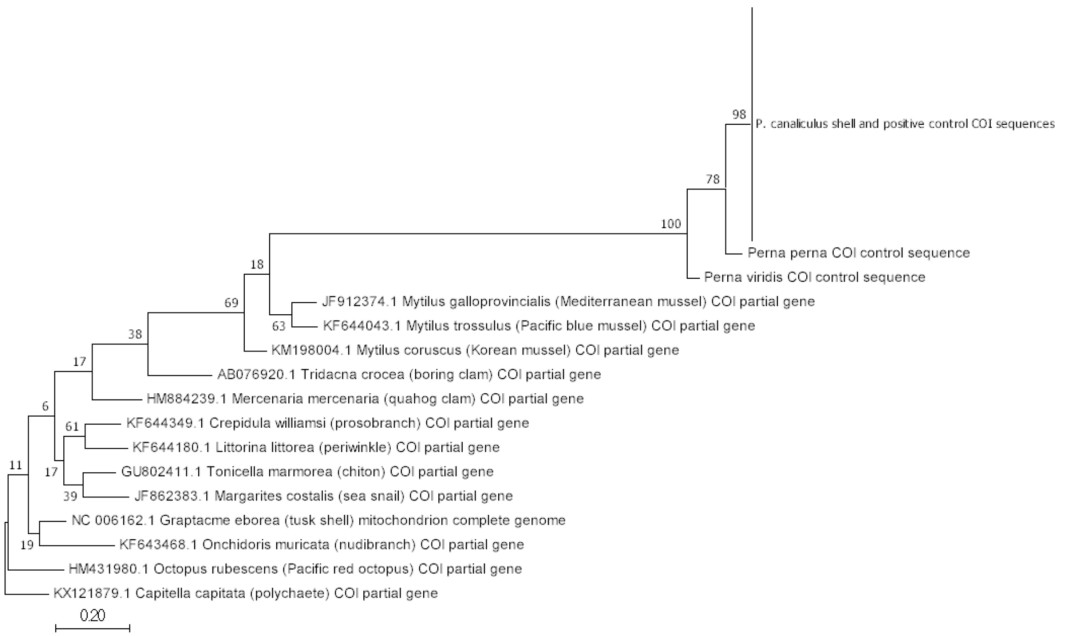

**Figure 3** **Molecular phylogenetic analysis of *P. canaliculus* COX1 shell sequences and positive control sequence, together with other COX1 mollusc species sequences.** Maximum Likelihood method based on the Tamura-Nei model (*Tamura & Nei, 1993*). Sequence KX121879.1 *Capitella capitata* was selected as outgroup. The tree with the highest log likelihood (−1371.62) is shown. The percentage of trees in which the associated taxa clustered together is shown next to the branches. Initial tree(s) for the heuristic search were obtained automatically by applying Neighbour-Join and BioNJ algorithms to a matrix of pairwise distances estimated using the Maximum Composite Likelihood (MCL) approach, and then selecting the topology with superior log likelihood value. The tree is drawn to scale, with branch lengths measured in the number of substitutions per site. The analysis involved a total of 130 nucleotide sequences. Codon positions included were 1st+2nd+3rd+Noncoding. All positions containing gaps and missing data were eliminated. There was a total of 110 positions in the final dataset. Evolutionary analyses were conducted in MEGA7 (*Kumar, Stecher & Tamura, 2016*).

sample with the lowest yield (0.01ng/µl or 0.002 ng DNA/mg shell) (Fig. 4). No statistical difference between the four treatments was detected when comparing Qubit measured DNA yields (one-way ANOVA, $F_{3,12} = 0.35$, $P = 0.79$, MS Excel (2016)).

Beach-cast shells had on average some three-fold higher Qubit measured DNA yields than cooked shells (Fig. 4). There was no significant difference (one-way ANOVA, $F_{2,21} = 1.95$, $P = 0.17$, MS Excel (2016)) found between the Qubit measured DNA yields from each of the three locations in the shells (ventral, palial sinus and umbo regions).

## DNA quality assessment by PCR

Successful PCR amplification for each primer set is depicted in Table 2. A total of 168 samples were tested with each of the mitochondrial primer sets. Pcan primers were used to successfully amplify, sequence and identify as *P. canaliculus* 85% of the samples. The COXI primer set was successful in 91% of the samples. The Chitin Synthase nuclear primer set was tested with a total of 80 samples, and successfully amplified, sequenced and identified as *P. canaliculus* in 40% of these.

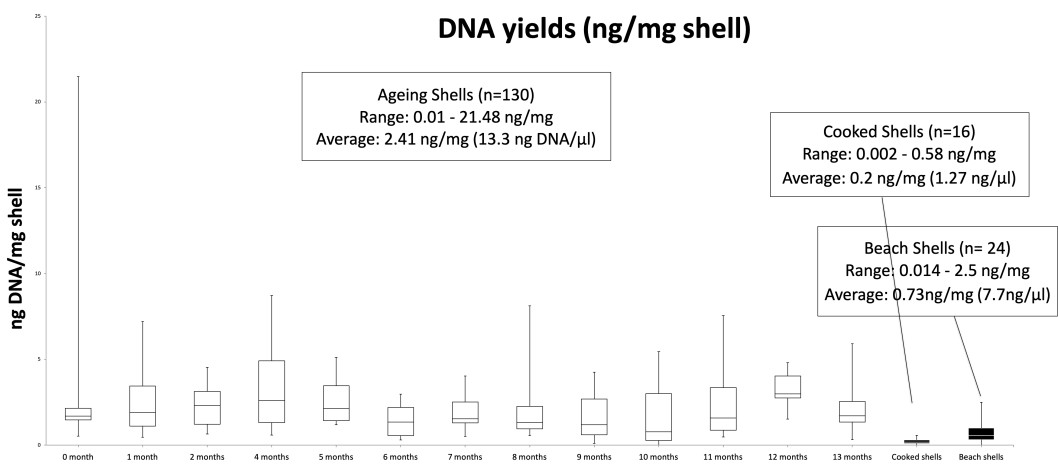

**Figure 4** **BoxPlot of average DNA yields.** Yields as ng DNA/mg shell weight ratio. Values through time in shell ageing experiment, and total average DNA yields from cooked and beach-cast shells (black filled box plot samples).

Between the three standard PCR primer sets used, a total of 416 reactions were tested, and of those, 253 (60.8%) were successfully amplified, sequenced and identified without template dilution. Template dilution in the remaining samples was helpful to a further 76 reactions, but had no effect in amplification success of the remaining 87 (20.9%). Most reactions that failed amplification despite template dilution used the chitin synthase nuclear primers. A total of 46 samples could not be amplified with this primer pair. Across all three primer pairs, the majority of failed reactions (52 samples) used beach-cast shell templates.

Elution of replicate DNA extracts in either TE buffer with low EDTA (0.1 mM) or deionized water did not improve amplification success.

Due to a combination of use of template in optimization trials, and evaporation in the microcentrifuge tube, two individual shell samples from the 5th month collection of the aged shell experiment did not have enough template for PCR analysis with any of our PCRs (Table 2).

Beach-cast shell samples, independently extracted in another laboratory, showed similar amplification successes to our own extractions, confirming the unlikelihood of source of contamination in our own research facility.

PCR using the Pcan primers did not amplify with either *P. viridis* or *P. perna* templates, confirming species-specificity for this assay. As expected, the COX1 PCR amplified equally well in all three *Perna* species.

## DNA quantification by qPCR

A total of 61 samples were tested with our qPCR assay (Table 2), with an average yield of 0.19 ng/μl for both ageing shells and cooked shells, and 0.02 ng/μl in beach-cast shells tested (Fig. 5). Yields in ng DNA/ mg shell were 4E–04, 3E–04 and 3E–05 for each of the treatments, respectively (Fig. 5). For every sample tested, qPCR determined DNA concentration (ng/μL) was lower than the corresponding Qubit measured concentration. However,
**Table 2  Samples successfully identified as *P. canaliculus* in each experiment.** Numbers in parenthesis relate to number of samples tested, and are followed by the number of samples that were successfully amplified.

| | | Mitochondrial Amplicons | | | Nuclear Amplicon |
|---|---|---|---|---|---|
| | Collection (month) | Pcan (305 bp) | qPCR Pcan (305 bp) | COX1 (191 bp) | Chitin Synthase (198 bp) |
| Aged shells | 0 ($n = 10$) | 8 | ($n = 10$) 9 | 9 | 8 |
| | 1 ($n = 10$) | 10 | ($n = 9$) 9 | 10 | 7 |
| | 2 ($n = 10$) | 9 | | 9 | – |
| | 4 ($n = 10$) | 7 | | 9 | – |
| | 5 ($n = 8$) | 8 | | 8 | – |
| | 6 ($n = 10$) | 10 | ($n = 9$) 9 | 10 | 1 |
| | 7 ($n = 10$) | 10 | | 10 | – |
| | 8 ($n = 10$) | 10 | | 10 | – |
| | 9 ($n = 10$) | 10 | | 10 | – |
| | 10 ($n = 10$) | 10 | | 10 | – |
| | 11 ($n = 10$) | 10 | | 10 | – |
| | 12 ($n = 10$) | 10 | | 10 | – |
| | 13 ($n = 10$) | 10 | ($n = 8$) 8 | 10 | 3 |
| | N total = 128 | Average= 95.3% | Average= 97.2% | Average= 97.7% | Average= 47.5% |

| | | Treatment | Pcan (305 bp) | qPCR Pcan (305 bp) | COX1 (191 bp) | Chitin Synthase (198 bp) |
|---|---|---|---|---|---|---|
| Cooked shells | Steam | 5 min ($n = 4$) | 4 | 4 | 4 | 3 |
| | | 10 min ($n = 4$) | 3 | ($n = 4$) 3 | 3 | 3 |
| | Embers | 5 min ($n = 4$) | 4 | 4 | 4 | 3 |
| | | 10 min ($n = 4$) | 4 | 4 | 4 | 3 |
| | | Average | 93.75% | 93.75% | 93.75% | 75% |

| | Shell region | Pcan (305 bp) | qPCR Pcan (305 bp) | COX1 (191 bp) | Chitin Synthase (198 bp) |
|---|---|---|---|---|---|
| Beach-cast Shells | Ventral ($n = 6$) | 2 | ($n = 4$) 3 | 3 | 1 |
| | Palial sinus ($n = 11$) | 2 | ($n = 2$) 0 | 7 | 0 |
| | Umbo ($n = 7$) | 2 | ($n = 3$) 2 | 3 | 0 |
| | Average | 25% | 55.5% | 54.2% | 4.2% |

qPCR determined yields in the aged shells showed a similar pattern of *P. canaliculus* DNA availability (Fig. 5) as the Qubit measured yields (Fig. 4) in the months that were tested.

qPCR was able to amplify three samples that had previously failed on all attempts of amplification using standard PCR with the same primers (Pcan primers) (Table 2).

## DNA amplicon sequencing and sequence analysis

BLAST results for amplicon sequences derived from the Pcan primer set had highest sequence similarity to GenBank Accession Ref. DQ343605.1 reporting the NADH4 mitochondrial gene for *P. canaliculus* (*Blair et al., 2006*) (average BLAST query cover (QC) and percent identity (Id) 89% and 98%, respectively). COX1 amplicon sequences

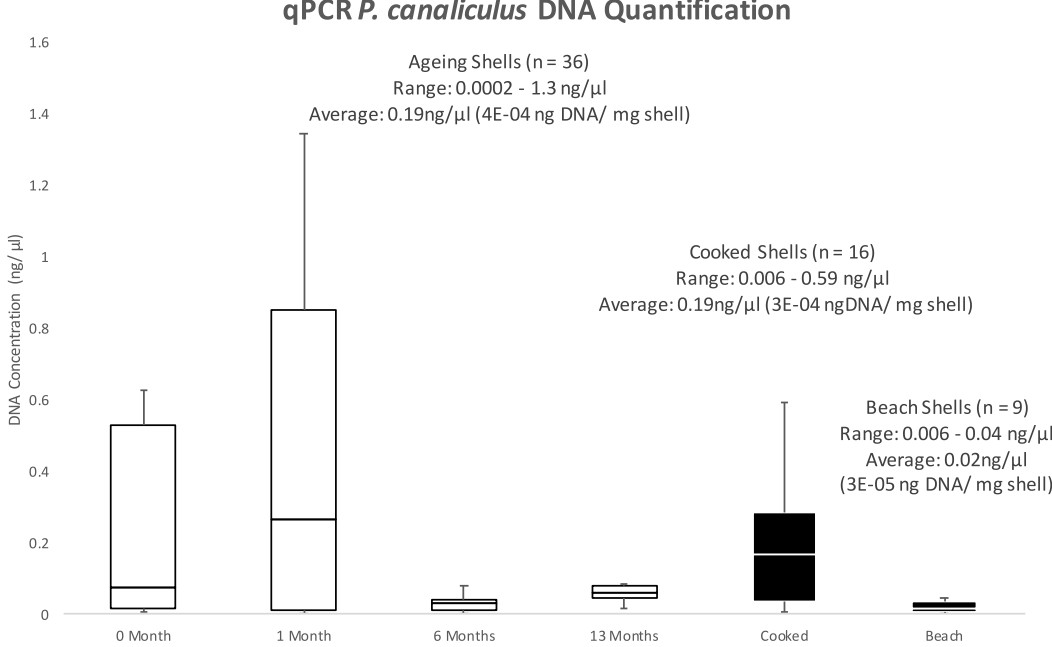

**Figure 5** **BoxPlot of DNA yields (ng/μl) determined by qPCR.** Showing the selected samples from the shell ageing, cooked shells and beach-cast shell collections.

generally aligned as most similar to either the *P. canaliculus* COX1 sequence with the GenBank Accession Ref. DQ343592.1 (*Blair et al., 2006*) (average BLAST QC 87% and Id 98%) or to GenBank Accession Ref. MG766134.1 (*Ranjard et al., 2018*), describing the *P. canaliculus* complete mitochondrion sequence (average BLAST QC 82% and Id 98%), with overall BLAST top results of QC 84% and Id 98%.

All chitin synthase amplicon sequences derived from our primers aligned solely with our GenBank ref. MT305043 for the *P. canaliculus* chitin synthase gene (partial, Fig. 2), with an average BLAST QC of 89% and Id of 99%.

## Aged shells

Mitochondrial amplification and subsequent identification by standard PCR of samples was 100% successful in all individuals of all but three collections (Table 2). Standard PCR using Pcan primers was overall slightly less successful than with using COX1 primers (95.3% amplification and identification compared to 97.7%) (Table 2).

Best results for the nuclear chitin synthase primer were achieved on samples from shells of freshly killed animals (month 0), and in the month 1 sampling (Table 2). The two other collections tested (6 and 13 months after death) had much lower rates of identification (10% and 30%, respectively) (Table 2).

qPCR with Pcan primers in the ageing shell sample subset successfully amplified all but one sample tested, including a sample on the 0 month collection that had not previously been amplified with standard PCR using the same primers (Table 2).

Both ng of DNA/ mg shell tissue ratio and qPCR calculated DNA yields show a similar pattern in the collections tested (months 0, 1, 6 and 13), in that there was an overall increase in yields from month 0 to month 1, followed by a decrease in the following months (Figs. 2 and 4).

## Cooked shells

93.8% of samples were identified with each of the mitochondrial primer sets, while the nuclear chitin synthase primers identified 75% of samples (Table 2). The sample with the lowest DNA yield of cooked shells, which was also the lowest value from all samples extracted for this work, failed to give results with all three amplicons with standard PCR, and also in the qPCR assay (Table 2).

To test our method even further, we collected and extracted replicate DNA samples from shell areas that were more highly charred in two different individuals in the five minute ember treatment (data not shown). There was no statistical difference detected (paired $T$-test, $p > 0.05$, MS Excel (2016)) in DNA yield values (ng DNA/ mg shell), however one of the charred samples failed to amplify with any of the PCRs, despite its uncharred paired replicate amplifying successfully in all of them (data not shown).

## Beach-cast shells

Despite higher Qubit measured DNA yields than the cooked shells (Fig. 4), this group of shell samples showed the lowest levels of successful amplification and identification with standard PCR. A total of 54.2% of samples were identified with the COX1 amplicon, and 25% with the Pcan amplicon overall (Table 2). Out of the 24 samples tested, only one amplified successfully and was sequenced with nuclear primers (Table 2). qPCR was successful in 55.5% of the subset of samples tested (Table 2), and calculated yields had an average of 0.02 ng/µl of *P. canaliculus* DNA in these samples (Fig. 5).

## DISCUSSION

Our results show that the carbonate shell matrix of *P. canaliculus* is a source of genetic material. The DNA yield from mussel shells likely originates from shell remodeling cells trapped and embedded in the matrix, which appears to provide good protection from enzymatic and chemical degradation that would otherwise degrade nucleic acid molecules in soft tissues. We have demonstrated that DNA can be extracted from *P. canaliculus* shells whether they are fresh, aged or cooked, and subsequently applied to amplify mitochondrial and nuclear regions of the genome.

Our DNA extraction method (*Gemmell & Akayama, 1996*) avoids the use of phenol-based solutions and extraction columns, and thus is a cheap, but effective alternative to commercial kits. Provided that the sample is at least 0.5 g in weight, and that primer design takes into account the need for short-length amplicons (*Taberlet, Waits & Luikart, 1999*), our data demonstrates the suitability of this method for extracting PCR-amplifiable DNA from bivalve shells for up to thirteen months, and in fresh samples prepared as they would be for human consumption.

We propose that the amount and quality of DNA that can be extracted and amplified from shell material relies on a balance between fine and coarsely grounded shell tissue,

and shell integrity, particularly of inner shell layers. We observed that disruption of shell samples of similar age showed variability of DNA yields (Fig. 4), as also observed by *Geist, Wunderlich & Kuehn (2008)* in fresh pearl mussel shells.

Well-weathered samples such as those from the beach-cast set had low shell integrity as they were very brittle, presented areas lacking the periostracum layer, breakage, and evidence of boring trematode parasites, likely due to a combination of age and environmental exposure causing DNA degradation, which translated as the group of samples with lowest DNA yields tested (Fig. 5). This relationship was also observed by *Der Sarkissian et al. (2017)* when extracting DNA from ancient shell samples. Loss of the periostracum layer during cooking of fresh shells decreased DNA extraction yields, but did not have an effect on DNA analysis success in this group of samples, even in those collected from charred areas.

qPCR DNA quantification showed a similar trend in the amounts of DNA extracted through time as with quantifying using a Qubit® 2.0 fluorometer. It also identified significant foreign DNA contamination in the beach-cast shell samples, which allied with the low yields of species-specific DNA, possibly highly degraded, explains the poorest amplification results of all sample groups. We were therefore not able to establish a connection between shell sample area (ventral, palial or umbo) and DNA extraction success.

Co-extraction of PCR inhibitors from samples collected from the environment, such as with forensic or ancient samples, is a well-known issue (*Alaeddini, 2012*). Dilution of DNA extracts with DNase-free water is a common method to circumvent PCR inhibition (*Kemp, Monroe & Smith, 2006*), and appears to have been moderately successful in our experiment.

Both mitochondrial primer sets were successfully used to amplify shell extracts. As with (*Frantzen et al., 1998*), our COX1 primers amplifying a fragment ∼100 bp shorter that the Pcan amplicon (Table 1), were the most successful.

A slightly lower amplification success with mitochondrial primers in the earliest collections of the aged shell experiment (Table 2) could be linked to the presence of foreign DNA from biofilm or shell parasite organisms in the sample's surface. This DNA would be expected to degrade with time, as it is not imbedded in the shell's tissue matrix.

The sharp drop in nuclear DNA identification success subsequent to the first month of sampling could indicate that reliable extraction and amplification of nuclear DNA from aged *Perna canaliculus* shell may require specialized protocols. Other studies amplifying mitochondrial and nuclear DNA from shell extracts also reported poorer results with nuclear amplicons (*Barsh & Murphy, 2008*; *Geist, Wunderlich & Kuehn, 2008*). The disparity between mitochondrial and nuclear results is likely a reflection of higher cell copy numbers of mitochondrial DNA molecules compared to nuclear molecules and the inherent protection afforded by a circular genome. If, as suggested by *Mount et al. (2004)*, DNA in shells has its origin in occasionally trapped cells, the higher copy number of mitochondrial DNA molecules would enable amplification of mitochondrial sequences, but not necessarily of nuclear sequences. *Geist, Wunderlich & Kuehn (2008)* reported 89% success genotyping fresh shells of freshwater pearl mussels with nine microsatellite markers

in. Our nuclear amplification success with fresh shells in the ageing experiment was 80%, which is very comparable.

Both nuclear and mitochondrial DNA analyses are feasible in shells of bivalves prepared as they would be for human consumption by exposure to high heat, possibly even from charred shell regions. This is of particular significance for the analysis of shell midden material, which is usually characterized by stratified deposits of consumed shelled animals such as bivalve molluscs, and indicates that DNA imbedded in bivalve shell matrix is well protected against degradation from high temperatures such as those used for cooking food.

Mitochondrial DNA extraction and amplification from mollusc shells has been reported in a variety of species, from fresh shells of freshwater pearl mussels (*Geist, Wunderlich & Kuehn, 2008*) and Pacific oyster (*Wang et al., 2012*), up to 50 year old snail shells (*Andree & López, 2013*; *Caldeira et al., 2004*; *Villanea, Parent & Kemp, 2016*), 70 year old abalone shells (*Hawk, 2010*) and even 125 year old oyster shells (*Barsh & Murphy, 2008*). In fact, *Der Sarkissian et al. (2017)* recently established that shell DNA extraction from a variety of marine mollusc species aged up to 7,000 years BP is possible and can be used to retrieve not only mollusc endogenous DNA, but also that of the natural microbiome, which could include pathogenic organisms, symbiotic to living molluscs. It is therefore very likely that mitochondrial DNA analysis can be successful in *P. canaliculus* shells for periods longer than the ones tested here.

Our findings could be significant in a number of areas, such as genotyping of historical collections, bioprospection of invasive species, and potentially shell midden research. In the greenshell mussel aquaculture industry, engraving the shell of animals with an identification number is a common technique (*Camara & Symonds, 2014*) that could be coupled with DNA extraction using the shell chips produced as a way of genotyping valuable aquaculture specimens, or endangered mollusc species, without the need to kill the animals.

Ancient shells DNA analysis techniques for shells such as those described by *Der Sarkissian et al. (2017)* are exciting new developments in this field, demonstrating the ability of mollusc shell matrices of entrapping and providing protection against nucleic acid degradation post-mortem, potentially for thousands of years.

Bivalve molluscs have been widely used for coastal monitoring anthropogenic aquatic contaminants such as heavy metals (*Goldberg et al., 1978*; *Azizi et al., 2018*), micro plastics (*Ward, Rosa & Shumway, 2019*) and human pathogen transmission (*Gyawali et al., 2019*; *Razafimahefa, Ludwig-Begall & Thiry, 2019*) together with seasonal ecotoxicological problems such as algal blooms (*Gibble, Peacock & Kudela, 2016*; *Hinder et al., 2011*). Their shells have also been used for almost four decades as a proxy to determine exposure to heavy metal concentrations in modern samples (*Yap et al., 2002*; *Koide, Lee & Goldberg, 1982*), historic (*Wing et al., 2019*) and archaeological samples (*Binkowski et al., 2019*). Along with being a source of genetical material not just of the individual itself, but pathogens and conditions that they've been exposed to makes mollusc shells a valuable resource to assess the environment in present and past times.

### Funding

This work was funded by the University of Otago, New Zealand. The funders had no role in study design, data collection and analysis, decision to publish, or preparation of the manuscript.

### Grant Disclosures

The following grant information was disclosed by the authors:
University of Otago, New Zealand.

### Competing Interests

The authors declare there are no competing interests.

### Author Contributions

- Sara Ferreira conceived and designed the experiments, performed the experiments, analyzed the data, prepared figures and/or tables, authored or reviewed drafts of the paper, and approved the final draft.
- Rachael Ashby and Kim Rutherford analyzed the data, authored or reviewed drafts of the paper, and approved the final draft.
- Gert-Jan Jeunen conceived and designed the experiments, analyzed the data, prepared figures and/or tables, authored or reviewed drafts of the paper, and approved the final draft.
- analyzed the data, authored or reviewed drafts of the paper, and approved the final draft.
- Catherine Collins performed the experiments, authored or reviewed drafts of the paper, and approved the final draft.
- Erica V. Todd analyzed the data, authored or reviewed drafts of the paper, and approved the final draft.
- Neil J. Gemmell conceived and designed the experiments, analyzed the data, authored or reviewed drafts of the paper, and approved the final draft.

### DNA Deposition

The following information was supplied regarding the deposition of DNA sequences:

The *Perna canaliculus* chitin synthase gene region used to design nuclear primers and amplify nuclear DNA from our samples is avaialble at GenBank: MT305043.

### Data Availability

DNA yields (ng DNA/ mg shell) for all experiments are available in File S1. qPCR calculated DNA yields (ng/ul) are available in File S2.

### Supplemental Information

Supplemental information for this article can be found online at http://dx.doi.org/10.7717/peerj.9420#supplemental-information.

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
