# Peer review of "DNA from mollusc shell: a valuable and underutilised substrate for genetic analyses"

_PeerJ, doi:10.7717/peerj.9420_

## Round 0.1 · original submission · Minor Revisions

I have now received assessments from three reviewers, all very positive - well done! Interestingly, two independent reviewers noted that the manuscript would benefit from some structural changes, in particular within the Discussion section (see Reviewers 2 and 3 comments). Other than that, the required changes are minimal.

I will be looking forward to your revised manuscript, along with a point-by-point response to the reviewers comments.

With warm regards,
Xavier

·

Basic reporting

This paper presents an intersting study on the use of New Zealand greenlip mussel for DNA-based studies. Different treatments to the shells were applied and the extractbale amounts of DNA as well as thri quality were assessed. The article is very clear and concise. It also nicely links the findings with results from previous studies. It might be useful to add specific hypotheses, but I also liked the clear presentation of only objectives as in the current version.

Experimental design

The article presents original research within the scope of the journal. The research question is well defined and the identified knowledge gap is being filled by a well-planned study design following high technical (e.g. verification of results in another laboratory) standards. All methods are described with sufficient detail and information to replicate.

Validity of the findings

There is no doubt about the validity of the findings which are in several cases similar to findings from a previous study on the use of mollusc shells for DNA based studies (Geist et al. 2008).

Additional comments

I would like to congratulate the authors for this nice paper. Based on my positive evaluation in the boxes on "basic reporting", "experimental design" and "validity of findings", I recommend publication of this study without the need for an additional round of review. Two really minor points that I spotted are that not all Figures are consistent in terms of reporting DNA concentrations (I think referring to ng per g shell material, possibly in addition to ng/DNA per µl) would be useful. Also, the "freshwater pearl oyster" mentioned a couple of times in the paper is actually a "freshwater pearl mussel".

Reviewer 2 ·

Basic reporting

This is very clearly written and presented with appropriate context and referencing provided.

Experimental design

Experimental design is thorough and authors are commended with the detail provided in this facet that has potential for this manuscript to become a primary source of literature for subsequent studies.

Validity of the findings

Findings are supported by robust statistical analysis and appropriate use of controls and other means of ensuring QA/QC

Additional comments

This manuscript in general is very well presented with clear findings that will potentially enable other researchers to utilise the methods. The main issue with this manuscript as presented is the format of the Discussion. This would require some significant rewriting as currently it is quite repetitive of the Results section and isn't a succinct summary and discussion of those results. This manuscript is primarily one of a method development and validation, therefore the Discussion can be relatively brief and be as a means to just reiterate briefly the overall findings and provide any further context to understanding those results. Then a section on applications for this methodology across other species of Molluscs is welcome. I would recommend a significant consolidation of this section.

Reviewer 3 ·

Basic reporting

no comment

Experimental design

no comment

Validity of the findings

no comment

Additional comments

This is an interesting research contribution to our knowledge of utility of mollusc shell to ge-netic analyses. The authors conducted a series of experiments aimed at assessing the possibility of identifying taxa, differently preserved mollusc shells. They considered using this knowledge for examination of evolutionary processes or reconstruction of paleoenvironmental conditions. The authors compared fresh, aged, beach-cast and cooked mollusc shell. They used commer-cially important bivalve: New Zealand endemic greenshell mussel, Perna canaliculus. They used PCR and sequence analyses of mitochondrial and nuclear gene fragments.
The authors showed very high success rate for mitochondrial DNA amplification from shell tissue extracted thirteen months after the animal’s death. The authors showed that the extracting DNA method they used and developed was suitable for mollusk shell.
All methods are clearly and in detail described. Only in line 144-146, the description is not clear.

A very large part of the work is a description of the methodology and conducted experiments, even in discussions. I suggest shortening the descriptions of experiments particularly in Discussion, or moving them to the Results section.
Generally, the paper is well written and I will recommend it for publication

---

## Round 0.2 · accepted · Accept

Dear Sara and co-authors,

I am delighted to accept your revised manuscript for publication in PeerJ. Please note that reviewer #1 has left one last comment for you to address, and I have found a number of minor corrections for your consideration to be incorporated at the proofs stage (see annotated pdf attached). In particular, your reference list needs attention.

Thank you for your nice contribution to the eDNA field!

With warm regards,
Xavier

·

Basic reporting

The authors have now addressed all of the comments and the only small error I spotted is that "in" needs to be deleted at the end of the phrase in line 399.

Experimental design

No changes compared to previous version.

Validity of the findings

No change in assessment compared to previous assessment.

Additional comments

Thanks for carefully revising the paper. Please still correct the small error in l.399 where the word "in" needs to be deleted.

Reviewer 2 ·

Basic reporting

No comment

Experimental design

No comment

Validity of the findings

No comment